# G-Local Attention Graph Pooling for Graph Classification

## Abstract

Graph pooling is an essential operation in Graph Neural Networks that reduces the size of an input graph while preserving its core structural properties. This compression operation improves the learned representation of the graph, yielding to a performance boost on downstream tasks. Existing pooling methods find a compressed representation considering the Global Topological Structures (e.g., cliques, stars, clusters) or Local information at node level (e.g., top-$k$ informative nodes). However, there is a lack of an effective graph pooling method that integrates both Global and Local properties of the graph. To this end, we propose a two-channel Global-Local Attention Pooling (GLA-Pool) layer that exploits the aforementioned graph properties, generating more robust graph representations. The GLA-Pool can be integrated into any GNN-based architectures. Further, we propose a smart data augmentation technique to enrich small-scale datasets. Exhaustive experiments on eight publicly available graph classification benchmarks, under standard metrics, show that GLA-Pool significantly outperforms thirteen state-of-the-art models on six datasets while being on par for the remaining two. The code will be available at this `link`.

## 1 Introduction

Graph Neural Networks (GNNs) have rapidly gained prominence as a powerful tool for learning node embeddings, excelling in graph-level and node-level tasks such as graph and node classification (Gao et al., 2022; Cai et al., 2021). Within the scope of graph classification, the graph pooling methods play a pivotal role in GNN architectures that map the set of nodes or subgraphs into a compact representation to capture a meaningful structure of the entire graph (Pang et al., 2021). Early methods in graph representation learning, also known as global graph poolings (e.g., MaxPool (Bai et al., 2019) or MeanPool (Simonovsky & Komodakis, 2017)), are the simplest approaches, which reduce the size of the input graph by performing a sum or average of all nodes to represent the graph. However, such simple aggregation operations ignore the hierarchical information in the graph and may lose feature information. Recently, several advanced hierarchical pooling methods, such as node cluster and node selection methods (Ye et al., 2023; Chen & Gel, 2023; Lu et al., 2022; Gao & Ji, 2019; Luzhnica et al., 2019), have been proposed to tackle the limitations of global pooling and obtain state-of-the-art performance.

Node-selection methods such as TopKPool, SAGPool, and Topological Pooling (Xu et al., 2022; Chen & Gel, 2023; Lee et al., 2019) select the most important nodes based on their node feature values or attention scores and drop other unnecessary nodes. However, these pooling methods consider only the graph's Local Topological Structure (LTS) during the pooling operation and ignore the graph's Global Topological Structure (GTS). LTS refers to the relationships between individual nodes, characteristics, and neighborhoods in the graph, while GTS extracts overall architecture and connectivity patterns that describe the entire network. On the other hand, the node cluster methods like DiffPool Ying et al. (2018) and CliquePool Luzhnica et al. (2019) group similar nodes into a single super node by exploiting their hierarchical structure. The Cluster-to-Node graph pooling method used the Attention Mechanism (AM) to select the most important nodes from the cluster to obtain a more reasonable representation of clusters (Ye et al., 2023). The node cluster methods mostly focus on capturing the GTS without considering the LTS by assigning the nodes to several coarse subgraphs. However, there is a need to develop an effective graph pooling method combining two types of graph pooling layers that improve the graph representation by extracting a graph's Lo-

cal and Global Topological Structure (LGTS). To address these limitations of existing graph pooling methods, this study proposes a graph pooling approach that can effectively capture the LGTS of the graph.

In this paper, we propose a novel graph pooling method named Global Local Attention Pooling (GLA-Pool). Specifically, we design a dual-channel approach that can effectively capture the LGTS and node features in graph pooling. The first channel leverages the idea of a clique that performs cluster pooling to extract GTS in a graph. The existing clique pooling method (Luzhnica et al., 2019) considered all cliques equally informative without considering node features. To address this, we introduce an AM that calculates an attention score for each clique to select the most informative ones. Furthermore, we recognize that every node within a selected clique does not contribute equally to generate a meaningful graph representation during the pooling operation. We argue that the most important nodes from all ranked cliques should be captured in graph pooling. Therefore, we develop a LocalPool layer using multi-attention in the second channel to identify the most important nodes from all ranked cliques of the first channel to capture the LTS. We summarized our contributions as follows:

- We introduce a novel pooling layer, named GLA-Pool, that integrates the global structural properties of the graph with the local node's properties.
- We experimentally prove that considering both sources of information (global and local) yields a better-learned representation. We consistently outperformed 13 state-of-the-art models on six diverse and challenging benchmarks.
- We introduced an attention mechanism to find the most relevant global structures (cliques).
- Additionally, we introduce a Node Augmentation Dropping (NDA) method to enrich small-scale datasets to improve the generalization capabilities of GNNs.

## 2  RELATED WORK

Related work includes existing graph pooling studies focusing on global and hierarchical techniques. Global pooling methods usually use different aggregation functions, such as sum and mean, to aggregate the features of all nodes that generate a single vector representation for the entire graph. For example, Vinyal et al. (Vinyals et al., 2015) proposed a Set2Set framework that uses the Long Short-Term Memory model to generate graph representation by identifying informative nodes. Authors (Zhang et al., 2018) developed a novel GNN layer to capture multi-level node features and a SortPool to sort these features to keep more graph information. However, global methods ignore the graph's hierarchical structural information and nodes' relative importance. The hierarchical pooling methods leverage LTS information of the graph as learning the representation of the nodes. Based on designing properties, the hierarchical methods can be classified into two main classes: node-selection and node-cluster.

Node selection methods reduce the graph size by selecting the most informative nodes based on their node features or attention scores. Hongyang et al. (Gao & Ji, 2019) developed a TopkPool that uses node scalar values on a trainable projection vector as the score of the node. SAGPool (Lee et al., 2019) further adopts the GNN layer to calculate node scores. Jinheon et al. (Xu et al., 2022) proposed Multistructure Attention Convolutional (MAC) pooling that incorporates dual-node scoring strategies to obtain the importance of nodes. Node selection pooling is considered to be more computationally efficient than cluster pooling. However, it is important that these pooling techniques merely consider the LTS of the graph. The node cluster pooling methods first group similar nodes into clusters by exploiting their hierarchical structure, then transfer each cluster into a single super node. DiffPool (Ying et al., 2018) uses the GNN layer output to learn a differentiable soft cluster assignment for nodes to extract a cluster. C2N-attention further improves the DiffPool using the AM to obtain more reasonable cluster representations by learning the importance of nodes within clusters. In (Luzhnica et al., 2019), the authors introduced a novel clique pooling to learn the GTS by extracting the maximal cliques from the graph. Quasi-CliquePool (Ali et al., 2023) further improves the CliquePool by capturing the overlapping nodes between two cliques using a replicator dynamical algorithm. Graph Multiset Transformer (GMT) pooling incorporates node structural dependencies with a multi-head attention mechanism to enhance the graph representation by identifying the interaction between nodes. Jinlong et al. (Du et al., 2021) design a Multi-Channel Graph

Pooling (MuchPool) that combines TopkPool and DiffPool methods to create a more comprehensive hierarchical representation of graphs. A recent study (Chen & Gel, 2023) introduced a novel Wit-TopoPool, which integrates a witness complex-based topological embedding mechanism with a global pooling layer. This approach aims to extract comprehensive and discriminative topological information from graphs.

However, the MuchPool method suffers from considerable computational complexity due to the simultaneous operation of three distinct channels to capture feature information, global and local structural information. Furthermore, the method's dependency on DiffPool for graph coarsening adds complexity. To further improve the performance of the MuchPool method, this study proposes a simple GLA-Pool approach based on a two-channel strategy. The unique strength of our method lies in its sequential processing of global and local structures. The first channel serves as a foundation by capturing GTS through an enhanced clique pooling mechanism augmented with attention features. This sets the stage for the second channel, which takes the output of the first channel as input to further capture LTS using a multi-attention strategy.

## 3 PROBLEM FORMULATION

This section introduces the problem formulation and mathematical notations used in this study. We represent the input graph as $G = (V, E)$, where $V$ denotes the nodes and $E$ shows the edges in the graph. The connection between nodes of G can be represented by an adjacency matrix $A \in \mathbb{R}^{N \times N}$, where $N = |V|$ is the number of nodes. The matrix $X \in \mathbb{R}^{N \times d}$ represents the node features matrix, where $d$ is the dimension of the feature space. The graph pooling operation changes the number of nodes at each pooling layer, thus, we further identify the graph at the $l$-th layer as $G'^l = (V'^l, E'^l)$. The node feature and adjacency matrices of the pooled graph represent as $X'^l \in \mathbb{R}^{N'^l \times f'^l}$ and $A'^l \in \mathbb{R}^{N'^l \times N'^l}$.

**Input:** Given a graph dataset with graph labels $D = \{(G_1, y_1), (G_2, y_2), ...\}$, number of graph pooling layers $L_p$, clique ratio $C_r$ (see sec. 4.2) and pooling ratio $P_r$ (see sec. 4.3).

**Output**: Predict the unknown label of the graph, $\hat{Y}$.

## 4 METHODOLOGY

In this section, we give a detailed explanation of our proposed GLA-Pool, which is shown in Fig. 1. The GLA-Pool performs graph pooling operations using two channels that capture different aspects of a graph, including LGTS and node features, and then aggregate the results of these channels to form a new pooled graph. Furthermore, we designed a node augmentation method in order to enrich the small datasets and enhance the generalization and robustness of our proposed pooling method. The following subsections present a detailed description of the proposed GLA-Pool.

### 4.1 NODE AUGMENTATION

In graph-based deep learning models such as GNNs, a significant challenge often arises due to the distinct attributes of graph data. This challenge is particularly evident when a comprehensive labelled dataset is unavailable for training. Hence, data augmentation techniques can effectively improve the generalization capabilities of GNNs by generating new data instances based on the existing dataset. However, the traditional data augmentation approaches, such as translation and rotation, are not directly applicable to graphs due to their complex topological structural information and node features. To this end, we design a simple and efficient NDA approach based on node degree: we randomly drop nodes having a small degree, generating new graphs. To this end, two parameters are introduced: the minimum degree (in our case = 2) and the drop probability (we set it to 0.15). See Algorithm 1 in appendix C for more details. This probabilistic approach serves a dual purpose: First, it enriches the diversity of the augmented graphs generated during training, and second, it improves the model's ability to generalize to unseen data. By selecting nodes with a lower degree, we ensure that the graph's core structural properties and patterns, often represented by high-degree or hub nodes, are preserved. So, by employing the node augmentation technique, we generate more reasonable augmented graphs, thereby enhancing our graph pooling method's robustness and generalization capability.

## 4.2 Channel 1: Global Topological Structure Learning

In channel 1, we aim to capture the GTS information of the graph. For this purpose, we use the modified version of the Bron-Kerbosch algorithm (Cazals & Karande, 2008), which is presented to enhance performance on large real-world graphs. So, this channel uses the Bron-Kerbosch algorithm to extract all cliques from the input graph $G$, denoted as a set $C_k = \{c_1, c_2, ..., c_k\}$. To handle nodes that belong to multiple cliques, we assign nodes to cliques based on two conditions: 1) If a node is already assigned to a clique, it is only added to the current clique being considered if the sizes of the existing and current cliques are equal and 2) If nodes of a clique have already been assigned to larger cliques, then we remove that clique. We also capture the overlapping nodes between two cliques. For instance, a node is connected to multiple cliques. In that case, we evaluate whether this specific node has maximum connectivity with the nodes of another clique. If it does and is not already part of the associated clique, the node is integrated into it. This process ensures that a node is assigned to the most relevant cliques, which refines the overall structure and improves the quality of the subsequent graph pooling step.

The existing clique pooling method (Luzhnica et al., 2019) partitions the graph into maximal cliques based on the graph's topological structure without considering the node features. Furthermore, this method treats all cliques equally, which can be problematic because not all cliques are equally significant; some are rich in information, and others might merely consist of a single node, thus carrying significantly less information to subsequent pooling layers. We incorporate an AM to address these issues and calculate important scores for each clique. We use a graph convolution layer to obtain attention scores. The calculation procedure is outlined as follows:

$$S_n = \sigma(\tilde{D}^{-1/2} \tilde{A} \tilde{D}^{-1/2} X W) \tag{1}$$

where $S_n$ represents the attention scores of each node and $\sigma(.)$ is a nonlinear activation function (softmax). The $\tilde{A} = A + I$ shows the adjacency matrix with self-loops, $\tilde{D} = \sum_j \tilde{A}_{ij}$ is the degree matrix of $\tilde{A}$ and $W$ is a trainable weight matrix. To calculate the score for each clique, various aggregation functions like average, max, and sum can be employed. In our implementation, we choose the sum function to aggregate the attention scores of the nodes within the corresponding clique $C_k$. Given a clique $C_k$, the clique score $S(C_k)$ is computed in equation 2. After obtaining the clique information score, we choose cliques with larger clique information scores in constructing the pooled graph because they can provide more informative graph sub-structures. In detail, we first sort the cliques based on their information scores, then select the top-ranked cliques as follows:

$$S(C_k) = \sum_{i \in C_k} S_n(i); \quad C_k' = rank(S(C_k), \lceil C_r * C \rceil) \tag{2}$$

where rank(.) denotes the function that returns the important cliques, $S(C_k)$ represents the clique scores, $C_r$ is the clique ratio, and $C$ is the total cliques. We update the node feature and adjacency matrix based on the nodes of selected cliques. This can be represented mathematically as:

$$X' = X \odot M \tag{3}$$

where $X'$ is the new node feature matrix, $X$ is the original node feature matrix, $\odot$ represents the broadcasted element-wise product, and $M$ is a vector where each entry corresponds to a node in the graph. The dimension of $M$ is the same as the number of nodes in the graph. Each entry in $M$ specifies whether or not the corresponding node is part of a selected clique (1 for nodes in $C_k'$ and 0 for all other nodes). This way, we can generate a new adjacency matrix where only the edges between the nodes in the selected cliques are retained. This can be represented as:

$$A' = A \odot M_e \tag{4}$$

where $A'$ is the new adjacency matrix and $A$ is the original edge index matrix. $\odot$ denotes element-wise multiplication, and $M_e$ is an edge mask (a boolean matrix of the same shape as $E$, where 1 indicates valid edges, and 0 indicates invalid ones. The first channel's output serves as an input for the subsequent channel. The information provided by this channel acts as a contextual framework that enables the second channel to identify local structures effectively.

## 4.3 Channel 2: Local Topological Structure Learning

This section presents how our proposed LocalPool layer leverages multi-attention mechanisms to learn LTS information by extracting the most informative nodes from ranked cliques. Existing

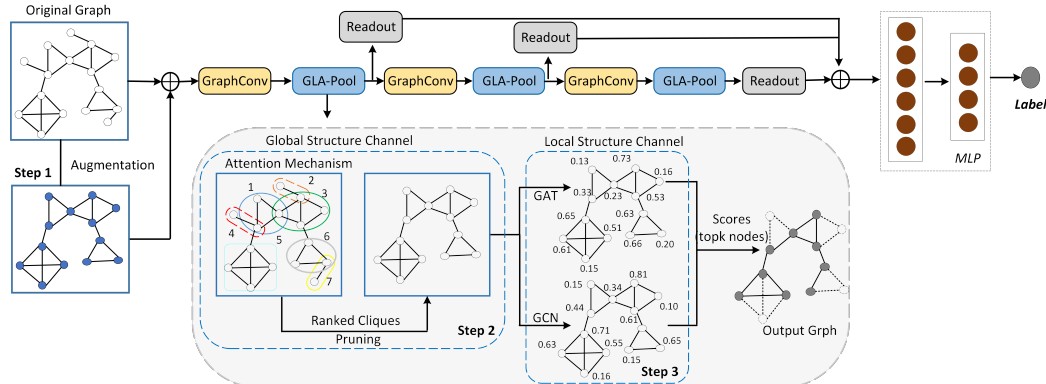

Figure 1: The hierarchical architecture of the proposed GLA-Pool integrates with GNN for graph classification. In the **first step**, the data augmentation method generates new graphs and combines them with existing data. The grey-lightbox demonstrates the overall workflow of our GLA-Pool method, which contains two channels. In the **second step**, the first channel performs three steps: 1) capture GTS by finding all maximal cliques within the graphs; 2) find overlapping nodes between two cliques to obtain more reasonable clique representations; and 3) incorporate the AM to select the most informative clique to form a new pooled graph. In the **third step**, the second channel takes this pooled graph as input and captures the LTS by selecting several significant nodes from ranked cliques based on scoring criteria (e.g., GCN and GAT). The readout function is applied after each pair of GNN and GLA-Pool layers to generate the graph-level representation.

pooling methods cliquePool and kplexPool (Luzhnica et al., 2019; Bacciu et al., 2021) extract all possible maximal cliques from the input graph and often transfer each clique into a single node to form a pooled graph. This transformation aims to reduce the graph size and complexity by aggregating the information within each clique. However, this transformation does result in a loss of information at the individual node level within each clique. The fine-grained details of interactions and relationships between nodes within a clique are not preserved in the pooled representation. This loss of information can potentially affect the downstream tasks, especially if the specific node-level information is crucial for classification or prediction tasks.

To address this, we propose a LocalPool layer to refine the graph further by selecting the most important nodes from ranked cliques. This refinement and selection process allows our pooling operation to help preserve the key information within each clique while reducing the graph size and complexity. Our LocalPool approach significantly diverges from existing methods like SAGPool (Lee et al., 2019). Specifically, SAGPool relies on a single strategy for computing node importance, yielding less robust node rankings. Inspired by the MAC method Xu et al. (2022), we adopt a dual-strategy-based pooling layer to calculate node importance, as illustrated in Figure 1. To generate attention scores for the nodes, we employ a GCN layer. This choice stems from the GCN model's ability to learn node representations by aggregating information from neighboring nodes, thereby offering a localized perspective. We use Equation 1 of GCN, along with the updated node feature and adjacency matrices, to calculate the scores of the nodes. Simultaneously, we employ the Graph Attention Network (GAT) layer to compute attention scores. This approach enables our pooling method to select nodes of utmost importance based on their individual features and their interrelations within the graph. Given that GCN and GAT extract graph information from distinct viewpoints, we consider this process as a multi-attention strategy for calculating node importance scores. The following outlines the computational process:

$$S = Aggregation\left(\sigma\Big(\frac{1}{h}\sum_{h=1}^{h}\sum_{j \in N(i)} a_{ij}{}^h W^h X'_j\Big), \sigma\Big(\tilde{D}^{-1/2}\tilde{A}'\tilde{D}^{-1/2}X'W'\Big)\right) \quad (5)$$

where $S$ represents an attention score for a node, and Aggregation is an operation such as max, mean, and sum. We use the max function to select the maximum value from each row (e.g., for each node, choose the maximum score from either the GCN or GAT model). The $h$ shows the number of heads, $a_{ij}$ is the attention weights between $x'_i$ and $x'_j$, $W^h$ is a trainable weight matrix of head $h$,

$X'$ is a feature vector, and $W'$ represents the trainable weight matrix of the pooled graph of the first channel. Utilizing the scores from $S$, we identify and select high-score nodes to construct a pooled graph. The details of this process are as follows:

$$idx = topk(S, \lceil P_r * N' \rceil); \quad X^{l+1} = X'(idx, :) \odot S(idx, :); \quad A^{l+1} = A'(idx, idx) \tag{6}$$

where $topk$ is the function that returns the indices of the top $N^{l+1} = \lceil P_r * N' \rceil$ values. The $X'(idx, :)$ perform the row-wise (node-wise) indexed node feature matrix, $S(idx, :)$ represents the row-wise indexed node importance score matrix, and $A'(idx, idx)$ is the row-wise and col-wise indexed adjacency matrix. $X^{l+1}$ and $A^{l+1}$ represent the new node feature and adjacency matrices, respectively.

## 4.4 THE READOUT FUNCTION

As shown in Figure 1, the hierarchical neural network architecture repeats the GNN and GLA-Pool operations many times so that we would observe multiple subgraphs with different sizes in each level $\{l_1, l_2, ..., l_k\}$. We use mean-pooling and max-pooling as a readout function to aggregate all the node representations within the subgraph to obtain a fixed-size graph representation as follows:

$$\Gamma_j = \left[ \max_{1 \leq i \leq N^l} X_{ij}^l \right] \forall j \in [0, d]; \quad R^l = \left[ \frac{1}{N^l} \sum_{i=1}^{N^l} x_i^l \quad || \quad \Gamma \right] \tag{7}$$

where $N^l$ is the number of nodes at layer $l$-th, $x_i$ is the feature vector of $i$-th node, and $||$ denotes concatenation. The $\Gamma \in \mathbb{R}^d$ vector contains the maximum values of each feature dimension across all nodes. The readout $R^l \in \mathbb{R}^{2*d}$ concatenates the two (average, and max) feature representations.

## 4.5 HIERARCHICAL POOLING ARCHITECTURE OF GLA-POOL

We employ a hierarchical pooling architecture, integrating multiple GCN layers with GLA-Pool layers to perform graph classification tasks. Figure 1 shows the architecture consisting of three blocks with a GCN and GLA-Pool layers. Each block receives a graph as input and generates a pooled graph with a new feature and adjacency matrices. Concurrently, a readout layer summarises the node embeddings into a single graph embedding. Finally, this graph embedding is subsequently fed into a multi-layer perceptron (MLP) classifier to make predictions regarding the labels.

## 5 EXPERIMENTS AND ANALYSIS

This section presents a comprehensive evaluation and quantitative analysis of our proposed GLA-Pool's effectiveness. We perform exhaustive experiments across three diverse dataset categories: chemical molecular structures (Riesen & Bunke, 2008; Wale et al., 2008) (including Mutagenicity, NCI-1, NCI-109, COX2, and BZR), social networks (Bacciu et al., 2021) (Reddit-B), and biological networks (Borgwardt et al., 2005; Dobson & Doig, 2003) (DD and Proteins). These datasets are benchmarks against which we compare GLA-Pool with state-of-the-art baseline methods. Additionally, we conduct ablation studies to evaluate the individual contributions of each channel within the GLA-Pool layer. To further analyse the robustness of our approach, we also explore the impact of hyperparameter variations on the performance of our pooling method.

### 5.1 BASELINES AND EXPERIMENTAL SETTINGS

**(1) Graph neural networks:** We select four GNN architectures to conduct comparative experiments: GCN (Kipf & Welling, 2016), GraphSAGE (Hamilton et al., 2017), GAT (Veličković et al., 2017) and GIN (Xu et al., 2018), which focus on learning node-level embeddings.

**(2) Graph pooling methods:** In this group, we compare our GLA-Pool with thirteen state-of-the-art hierarchical and global graph pooling approaches such as Set2Set (Vinyals et al., 2015), DiffPool (Ying et al., 2018), SortPool (Zhang et al., 2018), TopkPool (Gao & Ji, 2019), SAGPool (Lee et al., 2019), CliquePool (Luzhnica et al., 2019), GMT (Baek et al., 2021), MuchPool (Du et al., 2021), HoscPool (Duval & Malliaros, 2022), MAC (Xu et al., 2022), Quasi-CliquePool (Ali et al., 2023), C2N-ABDP (Ye et al., 2023) and Wit-TopoPool (Chen & Gel, 2023).

Table 1: Comparison of GLA-Pool and baselines on biological, chemical, and social networks benchmarks. The highest accuracy score is in **bold**, and the second highest score is in underline. A dash ('–') indicates the absence of publicly available records for the method on the dataset.

| Class | Baselines | PROTEINS | D&D | NCI-1 | NCI-109 | REDDIT-B | COX2 | BZR | Mutagenicity |
|---|---|---|---|---|---|---|---|---|---|
| GNNs | GCN[2017] | 74.75 ± 4.63 | 75.13 ± 4.14 | 79.68 ± 2.05 | 78.05 ± 1.59 | 71.05 ± 3.18 | 78.90 ± 2.74 | 79.34 ± 2.43 | 76.53 ± 1.82 |
| | GAT[2018] | 77.37 ± 2.95 | 72.65 ± 3.18 | 79.88 ± 0.88 | 79.92 ± 1.52 | 47.60 ± 2.96 | 81.70 ± 1.04 | 81.85 ± 1.57 | 79.85 ± 1.40 |
| | GraphSAGE[2018] | 76.73 ± 3.00 | 77.48 ± 3.20 | 79.98 ± 1.84 | 79.93 ± 1.52 | 81.70 ± 0.42 | 78.20 ± 0.02 | 78.21 ± 0.03 | 76.70 ± 0.20 |
| | GIN[2019] | 68.17 ± 2.39 | 65.94 ± 1.87 | 57.49 ± 0.73 | 56.62 ± 0.61 | 80.01 ± 0.26 | 80.30 ± 5.17 | 85.60 ± 2.00 | 81.70 ± 0.14 |
| Pooling | Set2Set[2015] | 70.26 ± 4.06 | 64.00 ± 6.82 | 68.95 ± 2.51 | 66.37 ± 6.18 | 83.60 ± 0.21 | 79.20 ± 0.02 | 78.21 ± 0.04 | 78.00 ± 0.19 |
| | DiffPool[2018] | 77.62 ± 4.97 | 80.99 ± 2.98 | 80.36 ± 1.56 | 78.51 ± 1.20 | 79.90 ± 0.08 | 77.60 ± 2.70 | 83.93±4.41 | 79.66±2.64 |
| | SortPool[2018] | 74.66 ± 5.08 | 67.47 ± 6.23 | 70.58 ± 3.68 | 68.87 ± 2.38 | 77.20 ± 0.35 | 78.20 ± 0.02 | 79.70 ± 0.07 | 75.80 ± 2.43 |
| | TopkPool[2019] | 78.26 ± 4.52 | 80.51 ± 2.17 | 69.73 ± 1.91 | 70.25 ± 2.09 | 78.60 ± 4.30 | 75.91 ± 3.60 | 79.40 ± 1.20 | 80.30 ± 4.21 |
| | SAGPool[2019] | 71.86 ± 0.97 | 76.45 ± 0.97 | 74.18 ± 1.20 | 74.06 ± 0.78 | 81.70 ± 2.20 | 78.30 ± 0.23 | 82.95 ± 4.91 | 79.45 ± 2.98 |
| | CliquePool [2019] | 73.86 ± 3.58 | 74.88 ± 4.35 | 78.83 ± 1.82 | 78.49 ± 1.14 | 81.02 ± 1.21 | 78.94 ± 3.22 | 82.17 ± 2.25 | 78.02 ± 1.12 |
| | GMT[2021] | 75.09 ± 0.59 | 78.72 ± 0.59 | 74.21 ± 1.88 | 71.38 ± 2.03 | 86.70 ± 2.60 | 58.90 ± 3.60 | - | 80.26 ± 2.20 |
| | MuchPool[2021] | 84.01 ± 1.73 | 84.80 ± 3.43 | 81.29 ± 1.31 | 80.50 ± 1.48 | 50.00 ± 0.01 | - | - | - |
| | HoscPool[2022] | 77.50 ± 2.23 | 79.40 ± 1.80 | 79.90 ± 1.71 | 78.50 ± 1.40 | 93.60 ± 0.09 | 64.60 ± 3.90 | - | 82.30 ± 1.3 |
| | MAC[2023] | 76.08 ± 3.55 | 79.13 ± 4.70 | 77.6 ± 1.66 | 75.84 ± 1.86 | - | - | - | 80.33 ± 1.49 |
| | Quasi-CliquePool[2023] | 78.68 ± 1.38 | 75.30 ± 3.30 | 80.11 ± 1.44 | 76.38 ± 2.20 | 82.55 ± 2.12 | 80.94 ± 3.12 | 83.55 ± 2.34 | 81.30 ± 1.23 |
| | C2N-ABDP[2023] | 79.00 ± 1.49 | 86.83 ± 1.15 | 80.77 ± 0.85 | 79.71 ± 0.75 | - | - | - | - |
| | Wit-TopoPool[2023] | 80.00 ± 3.22 | - | - | - | 92.82 ± 1.10 | 87.24 ± 3.15 | 87.80 ± 2.44 | - |
| Proposed | **GLA-Pool (our)** | **86.63 ± 1.47** | **87.16 ± 1.17** | **82.54 ± 1.10** | **81.13 ± 1.58** | 84.35 ± 3.58 | 85.66 ± 2.48 | **88.72 ± 2.12** | **86.27 ± 0.90** |

**(3) Experimental settings:** We implement our GLA-Pool method using the PyTorch framework (Paszke et al., 2017) and the PyTorch Geometric library (Fey & Lenssen, 2019). The maximal clique extraction component within the first channel is based on the CliquePool implementation (Luzhnica et al., 2019), which we modify to enhance clique representations by incorporating overlapping nodes between adjacent cliques. For a fair comparison, we follow many previous works (Gao & Ji, 2019; Lee et al., 2019; Ali et al., 2023; Du et al., 2021), employing tenfold cross-validation over 20 random seeds and reporting average accuracy along with standard deviation. Each dataset is divided into 80% training, 10% validation, and 10% testing subsets. Hyperparameters are optimized within predefined ranges, including embedding dimensions, learning rate, data augmentation ratio, batch size, pooling, and clique ratios (see the appendix section A.2). The Adam optimizer is employed for our model initialization, and a negative log-likelihood loss function is used for training. Our GLA-Pool model is trained for 200 epochs for all datasets.

## 5.2 PERFORMANCE COMPARISON

Table 1 outlines a performance comparison of the proposed GLA-Pool and other baseline methods based on accuracy and standard deviation across the eight benchmark datasets for the graph classification task. In this comparison, our GLA-Pool method achieves superior performance among all other baselines in the biological and chemical molecules domains datasets except COX2, in which we are second best. For instance, in the case of the chemical molecular domain, our GLA-Pool outperforms the best baselines by 1.25% improvement for the NCI1, 0.81% for the NCI109, 0.92% for the BZR, and 4.97% for the Mutagenicity. It is worth noting that, according to Table 3 in the Appendix, the average number of edges in these datasets is much smaller compared to the other datasets. This implies that these four chemical datasets are relatively sparse, presenting a significant challenge for graph pooling layers to learn meaningful representations. However, our designed dual-channel attention strategy can capture LGTS information of the graph, which enables GLA-Pool to extract information undeterred by the sparse graph structure effectively.

Furthermore, GLA-Pool consistently outperforms GCN-based global pooling methods across datasets from all three domains. This superior performance underscores the efficacy of our approach in generating more meaningful graph representations, thereby emphasizing the value of incorporating hierarchical pooling layers into the learning process. Notably, GLA-Pool and other node clustering techniques like CliquePool and DiffPool exhibit better performance than other GNN-based models such as GCN, GAT, GIN, and GraphSAGE. This outcome suggests that capturing Global Topological Structures (GTSs) in the form of maximal cliques or clusters is beneficial for enhancing graph representation learning. It is also noteworthy that CliquePool and DiffPool do not consistently outperform node selection pooling approaches such as SAGPool and TopkPool. This observation further substantiates the idea that integrating both node features and local and global topological information leads to more effective graph pooling methods. We can also see from Table 1 that our proposed GLA-Pool demonstrates considerable improvement in the biological domain datasets, with an

increase of 2.62% and 0.33% on Proteins and DD, respectively. Our analysis identified a significant proportion of isolated nodes and subgraphs in the Reddit-binary dataset. Furthermore, the graphs in this dataset demonstrate a scarcity of structures resembling maximal cliques. The effectiveness of our approach hinges significantly on the identification and ranking of such cliques, leading to challenges in capturing meaningful hierarchical representation structures within these graphs. This structural difference between the dataset and our method's underlying assumption likely contributes to the observed performance decline. We find that GLA-Pool consistently performs better on six out of eight benchmarks than baseline pooling techniques.

## 5.3 ABLATION STUDY

In this subsection, we conduct an ablation study on GLA-Pool by removing both channels to verify further where the performance improvement comes from. For convenience, we name the GLA-Pool method without the first channel (GTS) and the second channel (LTS) as GLAPool-NC1 and GLAPool-NC2, respectively. For these experiments, we chose four different-scale graph datasets (Proteins, NCI1, NCI109, and Mutagenicity). From Table 2, We can see that capturing GTS is crucial in chemical molecule graphs. Since the GLAPool-NC2 variant obtains a notable performance enhancement as graph attention facilitates the selection of the most informative cliques. The GLAPool-NC1 variant outperforms the GLAPool-NC2 variant in the biological domain dataset, while GLAPool-NC2 achieves better results in the chemical domain datasets (NCI1 and Mutagenicity). This pattern suggests that the LTS with node features are more important for some datasets than the GTS. Overall, GLA-Pool's ability to learn LGTS enables it to generate robust graph representations that significantly enhance performance in classification tasks.

Table 2: Effect of dual channels in GLA-Pool.

| Architecture | PROTEINS | NCI-1 | NCI-109 | Mutagen |
|---|---|---|---|---|
| GLAPool-NC1 | 79.2 ± 1.58 | 77.35 ± 1.89 | 77.01 ± 2.09 | 80.34 ± 1.27 |
| GLAPool-NC2 | 74.21 ± 3.99 | 79.88 ± 1.68 | 76.48 ± 2.24 | 81.31 ± 2.62 |
| GLA-Pool | **86.63** ± 1.47 | **82.54** ± 1.10 | **81.13** ± 1.58 | **86.27** ± 0.90 |

## 5.4 PARAMETER SENSITIVITY ANALYSIS

In this subsection, we investigate the sensitivities of the three main parameters influencing the performance of our GLA-Pool method: the pooling ratio $P_r$, the clique ratio $C_r$ and the node degree $ND_r$ ratio to drop nodes in NDA. Figure 4 in the appendix summarizes the accuracy of GLA-Pool under different combinations of parameters and is applied to three datasets, namely Proteins, NCI1, and Mutagen. The results indicated that the parameter values influence our model performance differently. For example, GLA-Pool achieves the highest accuracy on all three datasets when $P_r$=0.8 and $C_r = 0.8$. However, in a setting of $P_r = 0.5$ and $C_r = 0.5$, the GLA-Pool has the highest accuracy on Mutagen and Proteins, while the opposite result is achieved on NCI1. Notably, the accuracy decreases when we increase the NDr, especially in NCI1 and Mutagen. This performance degradation is likely due to losing the graph's original topological structures when several nodes are removed. Interestingly, the GLA-Pool demonstrated stability, producing reasonably satisfactory results across most combinations of these parameters.

## 5.5 GRAPH VISUALIZATION

To further demonstrate the distinctiveness and superiority of our pooling method compared to existing techniques, we use the Networkx library to visualize the pooling outcomes of GLA-Pool, CliquePool, TopkPool, and MuchPool. To provide a fair comparison, we build a hierarchical pooling architecture consisting of two layers and set a 0.8 for the $C_r$ and $P_r$. We randomly selected a graph from the Mutagenicity dataset comprising 36 nodes for the demonstration. The input and pooled graphs of the first and second pooling layers for each method are shown in the first row and the second row of Figure 2, respectively. The results demonstrate that the GLA-Pool, CliquePool and MuchPool mostly preserved the input graph's significant topological structure (ring and branch structures). In contrast, the results of the TopKPool are scattered with numerous isolated nodes, indicating a lack of structural preservation. The results obtained from the second pooling layer indicate that three baselines encounter challenges with preserving the underlying topology of the initial graph. However, GLA-Pool has the ability to preserve reasonable topological structures, such as dual ring structures present in the initial graph. This underscores the effectiveness of GLA-Pool since ring structures are crucial in the characterization of molecules.

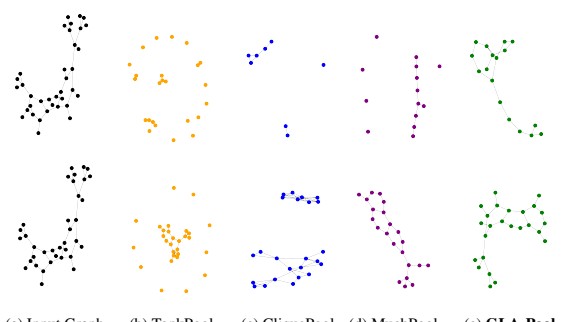

(a) Input Graph   (b) TopkPool   (c) CliquePool   (d) MuchPool   (e) **GLA-Pool**

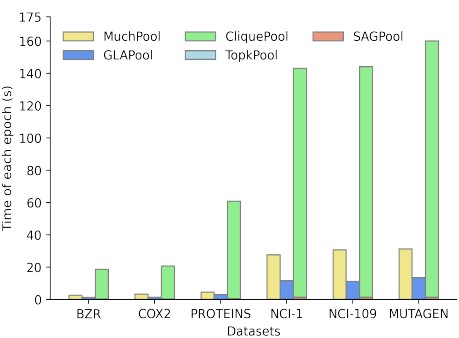

Figure 2: Comparison of various pooling methods in terms of graph visualization of different pooling layers.

Figure 3: Time efficiency of GLA and other baselines.

### 5.6 TIME EFFICIENCY

This subsection conducts a comparative analysis of our approach and several baseline methods in terms of time complexity. We select five varying scale graph datasets, including BZR, COX2, NCI1, NCI109, and Mutagenicity. After executing an identical number of epochs (200) for all models, we recorded each epoch and the total training time. We set a 0.5 pooling ratio and 128 batch size for all models to ensure a fair comparison. We trained all models on an NVIDIA RTX5000 GPU with 16GB of dedicated memory. As shown in Fig. 3, CliquePool and MuchPool's computation costs are relatively higher than other methods. Our method performs more efficiently than MuchPool and CliquePool. We achieve this by integrating an AM in the first channel to select only the most informative maximal cliques for pooling operations and eliminate unimportant ones. This approach significantly decreases the time cost of our first channel. However, the computational cost of GLA-Pool is comparatively more than TopKPool and SAGPool since our pooling method incorporates both LGTS information. So, it is a reasonable trade-off given GLA-Pool's superior performance.

### 5.7 LIMITATIONS

Our pooling method is based on the fundamental notion of clique. It is specifically designed to capture the LGTS information of the graph sequentially. However, a limitation arises when using the GLA-Pool on datasets where graphs do not necessarily exhibit structures like maximal cliques (e.g. many isolated nodes). For example, if our method fails to accurately capture a graph's representation in the form of maximal cliques at the first stage. In that case, further processes, such as ranking via the AM and implementing top-k pooling, would not provide optimum outcomes.

## 6 CONCLUSION

In this study, we introduced G-Local Attention Graph Pooling, a novel approach for acquiring hierarchical graph representations. Our method employs a two-channel strategy to sequentially capture both global and local topological structures within the graph. The first channel performed mainly three tasks: 1) identify all maximal cliques within the graphs; 2) improve the representation of these cliques by finding overlapping nodes between two cliques; and 3) incorporate the AM, which enables the selection of the most significant global topological structures as ranked maximal cliques. Then, we keep these important cliques and remove the unimportant ones to form a pooled graph. In the second channel, we designed a LocalPool layer using a multi-attention mechanism to obtain a robust node score. Next, we selected the most informative nodes based on node scores from all ranked maximal cliques, thereby capturing the LTS of the graph. In the last, we formed a new pooled graph by aggregating the results of the second channel. Additionally, we designed a node augmentation method to enrich the small-scale datasets. The effectiveness of our proposed method has been rigorously assessed through extensive experiments across eight graph classification datasets spanning three distinct domains. In future work, we aim to refine the clique pooling technique to capture a broader range of internal graph structures, thereby further boosting the performance in graph classification tasks.

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

## A  DATASET AND IMPLEMENTATION DETAILS

### A.1  DATASETS

To demonstrate the effectiveness of our proposed pooling layer in learning latent representations of the entire graph, we utilize eight different-scale benchmark datasets derived from three real-world domains such as chemical molecular, biological networks, and social networks. Table 3 summarizes the detailed statistics of these eight datasets.

- **Chemical molecular datasets:** in this domain, we utilized Mutagenicity (Riesen & Bunke, 2008), NCI109, and NCI (Wale et al., 2008). Mutagenicity is a drug chemical compound dataset that can be classified into two classes: mutagen and non-mutagen. Both NCI1 and NCI09 are datasets of chemical compounds with anti-cancer properties that are either active or inactive. We also add two small scale chemical datasets, such as COX2 and BZR in this category. These datasets are publicly available from the National Cancer Institute (NCI).

Table 3: Characteristics and Statistics of eight datasets.

| Classification | Datasets | Total Graph | Total Nodes | Total Edges | Classes |
|---|---|---|---|---|---|
| Biological | Proteins | 1,113 | 39.06 | 72.82 | 2 |
| | D&D | 1,178 | 284.32 | 715.66 | 2 |
| Chemical | NCI-1 | 4,110 | 29.87 | 32.30 | 2 |
| | NCI-109 | 4,127 | 29.68 | 32.13 | 2 |
| | Mutagenicity | 4,337 | 30.32 | 30.77 | 2 |
| | COX2 | 467 | 41.22 | 43.45 | 2 |
| | BZR | 405 | 35.75 | 38.36 | 2 |
| Social | REDDIT-B | 2,000 | 429.63 | 497.75 | 2 |

- **Biological networks datasets:** Proteins (Borgwardt et al., 2005) dataset contains protein structures that are classified as enzymes or non-enzymes. DD (Dobson & Doig, 2003) is also a dataset of protein structures, where each amino acid represents a node, and two nodes are connected by an edge based on spatial proximity. This dataset can be divided into two categories: enzymatic and non-enzymatic.

- **Social networks datasets:** REDDIT-BINARY (Bacciu et al., 2021) is a dataset of online discussions on Reddit, where nodes represent users and make an edge between two nodes if at least one responds to the other's comment.

## A.2 IMPLEMENTATION DETAILS

We implement our GLA-Pool method with the PyTorch (Paszke et al., 2017) framework and Py-Torch Geometric library on an NVIDIA Tu102 GPU. The part of maximal clique extraction in the first channel follows the implementation of CliquePool (Luzhnica et al., 2019), and we update this method to improve the clique representations by extracting the overlapping nodes between two cliques and AM to rank the cliques. For all experiments, we evaluate our model performance using tenfold cross-validation and split each dataset based on the conventionally setting into 80% training, 10% validation, and 10% test sets. We train our model for 200 epochs within each fold for all datasets and report the average accuracy on the test set and standard deviation. We search the dimension embedding in the range of $\{32, 64, 128, 256\}$, the learning rate is searched in $\{0.01, 0.001, 0.0001, 0.05, 0.005, 0.0005\}$ and pooling and clique ratio are searched in $\{0.5, 0.6, 0.7, 0.8, 0.9\}$. We train our model with the data augmentation ratio of 0.15 and node dropping degree of 2. We set 64 batch size for all datasets. We use the Adam optimizer to initialize our model, and the negative log-likelihood loss function is utilized to train our model. We consider the accuracy scores reported by the original papers for all baselines. If the baseline method results were missing, we used the original authors' code (if available) with the suggested hyperparameters setting. For a fair comparison, we follow many previous works (Xu et al., 2022; Ali et al., 2023; Du et al., 2021) and use the same graph neural network as a message-passing function for our method.

## B HYPERPARAMETER SENSITIVITY ANALYSIS

We investigate how the sensitivities of three hyperparameters—the pooling ratio $P_r$, the clique ratio $C_r$, and the node degree $ND_r$ ratio to remove nodes in NDA affect the performance of our GLA-Pool approach. Figure 6 illustrates the accuracy of GLA-Pool under various parameter combinations and applies to three datasets (Proteins, NCI1, and Mutagen). The findings show that the parameter values had a distinct impact on our model's performance. For instance, our method performs best on all three datasets when we set $P_r$=0.8 and $C_r = 0.8$. However, when $P_r = 0.5$ and $C_r = 0.5$, the GLA-Pool has the highest accuracy on Mutagenicity and Proteins, whereas NCI1 has the opposite result. Notably, the accuracy decreases when we increase the NDr, especially in NCI1 and Mutagenicity. This performance decreases because NCI1 and Mutagenicity datasets contain sparse graphs, and removing a substantial number of nodes from these datasets exacerbates the sparsity. This makes it even more challenging for our method to identify meaningful patterns or features within the data. Interestingly, the GLA-Pool demonstrated stability, producing reasonably satisfactory results across most combinations of these parameters.

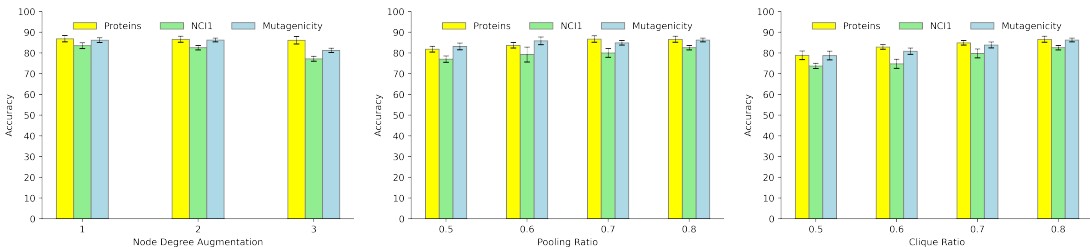

Figure 4: Hyperparameter analysis on GLA-Pool performance.

## C  NODE AUGMENTATION DROPPING METHOD

We introduce a Node Degree-based Augmentation (NDA) strategy that efficiently enhances the GNN's generalizability (see Algorithm 1). Our approach begins by calculating the degree of each node in the graph. Nodes with a degree less than or equal to a pre-defined threshold (in our case, 2) are then identified and probabilistically dropped from the graph with a set probability (0.15 in our experiments). This method serves two purposes: First, it increases the diversity of training data by generating varied graph structures, and second, it improves the model's capacity to generalize to unseen or novel data. Importantly, we preserve the graph's critical structural features by targeting low-degree nodes for removal, often represented by high-degree or 'hub' nodes. Consequently, the NDA technique generates a more diversified and representative set of augmented graphs, thereby bolstering the robustness and generalization capabilities of our graph pooling method.

---

**Algorithm 1** Node Augmentation

---

    **Input:** A graph $G$ with vertices $V$ and edges $E$, Threshold $T$ and Drop Probability $P$
    **Output:** Augmented graph $G'$
1: Initialize an empty set $D$ to store nodes to be dropped.
2: **for** each vertex $v$ in $V$ **do**
3:     Calculate node degree that $\deg(v) = |\{e \in E : v \in e\}|$
4:     **if** $deg(v) \leq T$ **and** random(0, 1) $< P$ **then**
5:         Add $v$ to $D$.
6:     **end if**
7: **end for**
8: Remove the nodes in $D$ from $V$ to get $V'$.
9: Updated the set of edges that $E' = \{e \in E : \nexists v \in D \text{ such that } v \in e\}$
10: Re-index nodes in $V'$ starting from 1 up to the size of $V'$.
11: **return** $G' = (V', E')$

---

# D GLA-POOL ALGORITHM FOR GRAPH CLASSIFICATION

The complete set of procedures for our proposed GLA-Pool method is comprehensively outlined in Algorithm 2.

---

**Algorithm 2** GLA-Pool Algorithm

---

    **Input:** Given a graph G as node features matrix X and Adjacency matrix A
    **Output:** The classification result as the label of graph G
1: **for** $i = 1$ to $l$ **do**
2:      Generate the augmented graphs using Algorithm 1
3:      Generate the node embedding X using the GCN layer;
4:      Calculate the attention scores of each node; $S_n = \sigma(\tilde{D}^{-1/2}\tilde{A}\tilde{D}^{-1/2}XW)$
5:      Calculate the clique's importance scores; $S(C_k) = \sum_{i \in C_k} S_n(i)$
6:      Select the most informative cliques; $C'_k = rank(S(C_k), \lceil C_r * C \rceil)$
7:      Update the node feature matrix $X'$ of graph G based on selected cliques; $X' = X \odot M$
8:      Update the adjacency matrix $A'$ of graph G based on selected cliques; $A' = A \odot M_e$
9:      Calculate multiple node importance scores for updated channel 1 output graph; $S =$
      $Aggregation\left(\sigma\left(\frac{1}{h}\sum_{h=1}^{h}\sum_{j \in N(i)} a_{ij}{}^h W^h X'_j\right), \sigma\left(\tilde{D}^{-1/2}\tilde{A}'\tilde{D}^{-1/2}X'W'\right)\right)$
10:    Node selection from the output graph (ranked cliques) of channel 1; $idx = topk(S, \lceil P_r * N' \rceil)$
11:    Update the pooled graph's node feature matrix; $X^{l+1} = X'(idx, :) \odot S(idx, :)$
12:    Update the pooled graph's node feature matrix; $A^{l+1} = A'(idx, idx)$
13: **end for**
14: Concatenate the node feature matrix of each pooled graph using the readout function; $\Gamma_j = \left[\max_{1 \le i \le N^l} X^l_{ij}\right] \forall j \in [0, d]; \quad R^l = \left[\frac{1}{N^l}\sum_{i=1}^{N^l} x^l_i \quad || \quad \Gamma\right]$

---

