# OpenReview forum: "G-Local Attention Graph Pooling for Graph Classification"
_ICLR.cc/2024/Conference — Submitted to ICLR 2024_

### Official Review · Reviewer_mksj · 2023-10-23

**Soundness:** 2 fair
**Presentation:** 2 fair
**Contribution:** 2 fair
**Rating:** 3
**Confidence:** 5

**Summary:**

This paper investigates graph pooling techniques in graph neural networks for graph classification task. Existing methods either use node clustering or node selection to reduce the size of the graph. In this paper, the authors propose GLA-Pool to incorporate both global and local information of the graph. Specifically, clique algorithm is utilized to extract all the possible maximal cliques, then each clique is transformed into a single node to form a pooled graph, which captures the global property. To capture its local property, an attention mechanism is performed in each clique to select important nodes. Experimental results on several public datasets and methods demonstrate that the proposed model can achieve satisfied performance.

**Strengths:**

1.	This paper studies graph pooling for graph classification, which is an important topic in graph neural networks.
2.	Different types of datasets are utilized to evaluate the model’s performance.
3.	Ablation studies are given to show the effectiveness of the proposed components.
4.	Visual figures are given to help the readers to understand the model.

**Weaknesses:**

1.	The novelty of the proposed model is limited since it simply combines CliquePool and SAGPool. There are almost no key modifications in the modules.
2.	The used datasets are too small and all the datasets are binary classification task. More large-scale datasets are suggested like ogbg-molpcba and ogbg-ppa.
3.	The experimental settings are not consistent. In Table 1, the authors directly cited the results from existing methods. However, their settings are not same and directly using their results are not fair. For instance, in MuchPool [2021], it used 10-fold cross validation. In Wit-TopoPool [2023], it utilized 90/10 random training/test split. In this paper, the authors use 10-fold cross validation with 80% training, 10% validation and 10% testing. Therefore, reproducing the results under same setting is suggested.
4.	It is not clear why GAT and GIN achieve such a poor performance in some of the datasets. For instance, GAT is 47.6 in Reddit-B and GIN is 57.49 in NCI-1. More discussions are encouraged in these special scenarios.
5.	In Figure 3, the proposed GLAPool has a lower time complexity compared with CliquePool. Is the time of maximal clique extraction included?

**Questions:**

1.	It is not clear whether the baselines are also using node augmentation in the training procedure.
2.	The motivation of using GCN and GAT as two views is not clear. What if we only use one of them?
3.	In Eq. (6), $S(idx, :) \in R^{N^{l+1} \times 1}$ cannot element-wisely multiply with $X^{'}(idx, :) \in R^{N^{l+1} \times d}$. There should be some transformation operations on $S(idx, :)$.
4.	It is not clear which GNN backbones are used in the experimental results. Although the authors claim that any backbone is applicable, there is not experimental results for support.

---

### Official Review · Reviewer_Cm8z · 2023-10-29

**Soundness:** 2 fair
**Presentation:** 3 good
**Contribution:** 2 fair
**Rating:** 3
**Confidence:** 5

**Summary:**

This paper introduces a method called GLA-Pool, which learns pooled graphs from both local and global perspectives. Extensive experiments have been conducted on the pooling operation to verify its effectiveness.

**Strengths:**

The consideration of both local and global information when designing the pooling operations is a significant aspect of this study.

**Weaknesses:**

1. The limited literature review results in a weak contribution to the field. The main challenge in this paper appears to be the design of the local and global structure learning components. For methods that incorporate global information, such as clique, cluster, and stars, there is a lack of comparison with these methods, leading to a weak justification for the first contribution. The same issue arises with methods for learning local information. Moreover, method [1] also focuses on capturing global structures, and [2] provides a detailed discussion on pooling operations. A comparison between existing methods and the two components designed in this study should be provided to justify their effectiveness.

2. The evaluation of data augmentations is overlooked. Although data augmentations are provided in this paper, their evaluations are ignored. It appears that GLA employs a data augmentation trick while the baselines do not, which creates an unfair advantage in the experiments.

3. The classification of LTS and GTS. This paper seems to categorize methods into two classes based on local and global topology extractions. It would be beneficial to explain how this differs from the selection and grouping-based methods mentioned in [2] and [3].



[1] Spectral clustering with graph neural networks for graph pooling. ICML 2020
[2] Understanding Pooling in Graph Neural Networks. TNNLS 2022
[3] Graph pooling for graph neural networks: Progress, challenges, and opportunities.

**Questions:**

Please check the weakness.

---

### Official Review · Reviewer_XDBL · 2023-10-30

**Soundness:** 2 fair
**Presentation:** 2 fair
**Contribution:** 1 poor
**Rating:** 3
**Confidence:** 4

**Summary:**

This paper introduces a two-channel attention-based graph pooling technique GLA-Pool that effectively incorporates both graph topology and node information into hierarchical graph pooling. The importance of graph pooling in GNNs is discussed. The authors conduct experiments on various datasets, demonstrating that GLA-Pool outperforms several existing GNNs and graph pooling methods.

**Strengths:**

1. The concept of integrating global topology and node information in graph pooling is straightforward and well-motivated.

2. The proposed method exhibits good performance on most datasets when compared to other graph pooling baselines.

**Weaknesses:**

1. The major concern on the paper is the lack of novelty. The paper appears to be an incremental amalgamation of existing works. In particular, the dual-strategy-based pooling resembles SAGPool in the way it generates attention with reference to clique information. The authors should provide a better positioning of their work in the existing literature.

2. The notation used in this paper lacks consistency and is confusing. Conventionally, bold capital letters are used to represent matrices, bold lowercase letters signify vectors, and lowercase letters denote scalars. However, the notation system in the paper mixes up these conventions: e.g., using "X," "M," and "C_r" to represent a matrix, vector, and scalar, respectively, making the equations hard to understand. Additionally, if "C" represents a set of total cliques, it should be denoted as "|C|" in Equation (2).

3. The paper lacks adequate discussions and comparisons with substructure-counting based methods, such as references [1], [2], and [3].

4. The experiments are conducted on small-scale datasets. It would be beneficial to include additional experimental results on large datasets, such as OGBG-MOLHIV and ZINC, to demonstrate the model's scalability and generalizability.

5. The ablation study is limited. Some aspects of the model design, such as node augmentation and the inclusion of GCN and GAT in the dual-channel, require further discussions and analyses.

6. The visualizations provided in the paper do not effectively support the motivation of using cliques in graph pooling. Given the limited presence of cliques with three or more nodes, it may be more informative to highlight the significance of capturing cycles in graph structures.

7. The presentation should be improved, especially for the methodology section. It would be helpful to polish the writing and incorporate illustrative figures or examples.

[1] "Uplifting any GNN with local structure awareness."
[2] "Improving graph neural network expressivity via subgraph isomorphism counting."
[3] "Boosting the cycle counting power of graph neural networks with I^2-GNNs."

**Questions:**

Please refer to Weaknesses. Some additional questions are:
- Why does the proposed method take the high-degree nodes as the core part of the graph in data augmentation? The authors treat nodes with low degrees unimportant and drop them. However, in applications such as molecular datasets with toxic/non-toxic compounds, the functional groups often contain low-degree nodes with benzene rings. The proposed method may not work in such applications.

- How is M_e generated? Is it based on the selected node or by another network?

---

### Official Review · Reviewer_iwUJ · 2023-11-01

**Soundness:** 2 fair
**Presentation:** 2 fair
**Contribution:** 2 fair
**Rating:** 3
**Confidence:** 4

**Summary:**

In this paper, the authors aim to propose a graph pooling method for enhancing the graph classification performance. In particular, the authors aim to capture both global and local properties of the graph for graph pooling. In addition, the authors also propose a data augmentation strategy to enrich small-scale datasets.

**Strengths:**

*Clarity*: In general, the paper is well-organized and easy to follow.

*Quality*: The paper conducted a set of experiments to verify the effectiveness of the proposed pooling method. The authors also conducted an ablation study to understand how different components help the model.

**Weaknesses:**

1. The novelty of this paper is somewhat limited. The proposed method seems to have very marginal contributions compared with existing solutions. For example, in Section 4.3, the local topology pooling operation is simply similar to SAGPool but with a combination of GCN and GAT for learning the importance score.

2. Some of the arguments are not very strict. For example, the authors propose to augment the graphs by alerting the nodes with low-degree. They claimed this is due to that the graph's core structural properties and patterns are often represented by high-degree nodes. It would be better if the authors could provide some references or investigations for such claims, especially for graph classification tasks.

3. Some of the model designs are not well-motivated. For example, it is argued that "SAGPool yields less robust node rankings due to a single strategy for calculating node importance". However, it is not very clear how combining GCN and GAT can help address this issue.


Minor Issues:

1. The combination of global and local information is not detailed in the main text of the paper. It is demonstrated in Figure 1. It might be better if the authors could provide some description for this part.

**Questions:**

Please answer the question listed in the weakness.

---

### Meta-Review · Area_Chair_94rH · 2023-12-06

**Metareview:**

This paper proposes a new graph pooling method, GLA-Pool, in graph neural networks for graph classification task. The GLA-Pool method transforms each clique into a single node, and uses an attention mechanism in each clique to select important nodes. Experiments on several public datasets and methods show satisfied performance.

The studied topic is an important one for graph neural network. The main idea is straightforward and well-motivated. Various types of datasets are utilized to evaluate the model's performance.

However, the novelty of this paper is very limited, since it is a simple combination of two existing methods. Secondly, the datasets are too small, and only are for binary classification. In addition, the experimental settings are not fair for all the compared methods. Finally, the paper should be further polished carefully.

**Justification For Why Not Higher Score:**

The major concern is the limited novelty.

**Justification For Why Not Lower Score:**

N/A

---

### Decision · Program_Chairs · 2024-01-16

Reject